# Defensive Tensorization: Randomized Tensor Parametrization for Robust Neural Networks

## Abstract

As deep neural networks become widely adopted for solving most problems in computer vision and audio-understanding, there are rising concerns about their potential vulnerability. In particular, they are very sensitive to adversarial attacks, which manipulate the input to alter models' predictions. Despite large bodies of work to address this issue, the problem remains open. In this paper, we propose defensive tensorization, a novel adversarial defence technique that leverages a latent high order factorization of the network. Randomization is applied in the latent subspace, therefore resulting in dense reconstructed weights, without the sparsity or perturbations typically induced by the randomization. Our approach can be easily integrated with any arbitrary neural architecture and combined with techniques like adversarial training. We empirically demonstrate the effectiveness of our approach on standard image classification benchmarks. We further validate the generalizability of our approach across domains and low-precision architectures by considering an audio classification task and binary networks. In all cases, we demonstrate superior performance compared to prior works in the target scenario.

## 1 Introduction

Deep neural networks (DNNs) are powerful predictive models that achieve impressive accuracy across a wide range of artificial intelligence tasks, including image classification (He et al., 2016; Krizhevsky & Hinton, 2009) and speech recognition (Amodei et al., 2016a; Graves et al., 2013; Graves & Jaitly, 2014). The popularity of DNNs in production-ready systems has raised a serious security concern as DNNs are found to be susceptible to a wide range of adversarial attacks (Madry et al., 2017; Akhtar & Mian, 2018; Dong et al., 2018; Huang et al., 2017; Goodfellow et al., 2014; Kurakin et al., 2016), where small and imperceptible perturbations of the input data lead to incorrect predictions by the networks. These shortcomings pose an obstacle in wide-scale adoption of DNNs and expose an inherent weakness in their reliability. This is especially important when such models become part of security and safety related solutions (Amodei et al., 2016b).

This susceptibility of DNNs to adversarial perturbations has led to a large volume of work that attempts to design robust networks (Dhillon et al., 2018; Lin et al., 2019; Samangouei et al., 2018; Song et al., 2017; Guo et al., 2017). However, advances in designing robust DNNs have been followed with stronger perturbation schemes that defeat such defences (Athalye et al., 2018).

Most defenses that rely on randomization, either apply randomized transformations to the input, e.g. (Xie et al., 2018), or randomization applied within the network, e.g., on the activations (Dhillon et al., 2018) or on the weights directly (Wang et al., 2018). However, all these approaches typically introduce artificats (e.g. sparsity in the weights or activations) and can be defeated by carefully crafted attacks (Athalye et al., 2018). In addition, it is preferable to not to rely on modifications of the inputs but have a network that is inherently robust against attacks. In this paper, we take a different approach to randomization. We first parametrize the network using tensor factorization, effectively introducing a latent subspace spanning the weights. We then apply randomization in that latent subspace, enabling us to create models that are robust to adversarial attacks, without modifying directly the weights, activations or inputs. In summary, we make the following contributions:

- We propose a novel adversarial defence technique that relies on a latent randomized tensor parametrization of each layer in the network and can be seamlessly integrated within any network architecture.

- We thoroughly evaluate the robustness of our method against various adversarial attacks and show that it consistently and significantly improves over the current state-of-the-art especially when combined with adversarial training.

- We show that our method successfully hardens the models against these attacks for both quantized and real-valued nets.

- We verify that our strategy works across domains, by experimenting on both image-based and audio-based classification.

## 2 RELATED WORK

In this Section, we review the related work on adversarial attacks (Section 2.1) and defences (Section 2.2), followed by an overview of tensor methods for deep learning (Section 2.3).

### 2.1 ADVERSARIAL ATTACKS

First, we review a few of the most popular adversarial attacks alongside the current defense strategies employed. Given a data sample, e.g., an image $\mathcal{X}$, an adversary will try to find a *small* perturbation, often *imperceptible* to a human eye, but that, added to the input sample, will cause it to be misclassified by the target model, with high confidence. Mathematically, the attacked generates a perturbation $\Delta$ bounded in terms of some $\ell_p$ norm, i.e. $\|\Delta\|_p \leq \epsilon$, typically with $p = 2$ or $p = \infty$. The adversarial sample is obtained by adding the perturbation to the input sample $\mathcal{X}_{adv} = \mathcal{X} + \Delta$.

Several ways of obtaining this adversarial perturbations have been proposed (Goodfellow et al., 2014; Athalye et al., 2018; Carlini & Wagner, 2017). Among them, *black box* attacks, consider the network as a black-box in which the attacker has no information regarding its architecture or the gradients. *White-box* attacks on the contrary, assume complete access to the network architecture and all its parameters. Moreover, attacks can either be *untargeted*, in which case the goal is simply to make the network predict *any* wrong label, or *targeted*, in which case the aim is to force the network to predict a specific label, independently from the input sample. Next, we introduce the main white-box attacks used in this paper.

**Fast Gradient Sign Method (FGSM)** is a single-step, gradient based technique, introduced by Goodfellow et al. (2014) to generate $\ell_\infty$-bounded adversarial perturbations as follows:

$$\mathcal{X}_{adv} = \mathbf{X} + \epsilon \cdot \text{sgn}(\nabla_{\mathcal{X}} \mathcal{L}(\theta, \mathcal{X}, y)) \tag{1}$$

where $\theta$ is the parameters of the target neural network. While the single gradient-step nature of FGSM makes it better for transferability attacks, this can also lead to a suboptimal ascent direction.

**Basic Iterative Method (BIM) and Projected Gradient Descend (PGD)** aims to address the shortcoming of FGSM by running it for several iterations. Kurakin et al. (2016) propose BIM, in which the FGSM is run for for several iteration, clipping the values of the perturbation at each step to be inside the bounds. Madry et al. (2017) further improve upon this by prepending BIM with a random start and replacing clipping with a projection onto the acceptable set:

$$\mathcal{X}_{adv}^{t+1} = \Pi_{\mathcal{X}+S}(\mathcal{X}_{adv}^t + \alpha \cdot \text{sgn}\left(\nabla_{\mathcal{X}} \mathcal{L}(\theta, X_{adv}^t, y)\right), \tag{2}$$

where $\alpha$ is the step size and $\Pi_{\mathcal{X}+S}$ is a projection operation forcing the generated adversarial samples to be in the $\ell_p$ ball $S$ around $\mathbf{X}$. A model resilient to PGD attacks is considered to be reasonably resistant to all first order attacks (Madry et al., 2017).

### 2.2 ADVERSARIAL DEFENCES

Despite recent advances, developing robust neural networks remains an open, challenging problem (Athalye et al., 2018). Current defense strategies typically attempt to either detect the adversarial samples and denoise them, or inject adversarial samples during training. The latter is known as adversarial training (Goodfellow et al., 2014; Kurakin et al., 2016; Madry et al., 2017) and is considered the most resilient defense technique. However, the above mentioned defences typically do not increase the robustness to black box attacks. In addition they can typically be defeated using

two-step approaches (Tramèr et al., 2017). Feature squeezing is another model hardening approach introduced by Xu et al. (2017). This technique proposes to reduce the complexity of the data representation which in turn causes the adversarial perturbations to disappear due to lower sensitivity. Guo et al. (2017) proposes a set of five transformation, namely *image cropping and re-scaling, bit-depth reduction, JPEG compression, pixel dropping and image quilting*, that applied to the image to increase the robustness to adversarial attacks of a given model. However, even the combination of all these transformations was shown to be vulnerable to carefully tuned attacks (Athalye et al., 2018). Samangouei et al. (2018) introduces the so-called *Defense-GAN* technique. The main idea is to project the samples into the manifold of a generator before classifying them. Similarly, Song et al. (2017) uses a PixelCNN instead of a generative model. Despite the variety of recently proposed defence strategies, in (Athalye et al., 2018) the authors show that most of the existing defense techniques rely on one form of gradient obfuscation (gradients shattering, stochastic gradients and vanishing/exploding gradients), proposing both a method to detect such class of defences and to defeat them.

As opposed to all the aforementioned works which either manipulate the data samples (Samangouei et al., 2018; Song et al., 2017; Guo et al., 2017) or introduce stochasticity on the activations (Dhillon et al., 2018) or weights Wang et al. (2018) of each layer, we propose a novel defense strategy that leverages tensor factorization of the weights in order to apply randomization in that latent space, before reconstructing the weights the weights. The approach is introduced in details in section 3.

### 2.3 TENSOR METHODS IN DEEP LEARNING AND RANDOMIZED TENSOR DECOMPOSITION

Tensors are high dimensional generalizations of matrices (Kolda & Bader, 2009). Recently, tensor decompositions have found a surge of applications in deep learning, mainly focusing on networks compression and acceleration. By parametrizing layers of neural networks using tensor decomposition, or even whole networks (Kossaifi et al., 2019a), the number of parameters can be reduced with little to no loss of performance, and in some cases the operations can be done more efficiently (Lebedev et al., 2015; Novikov et al., 2015; Kim et al., 2016; Astrid & Lee, 2017).

In some cases, tensor decompositions can exhibit high computational cost and possibly low convergence rates when applied to massive data. To accelerate computation, and enable them to scale, several randomized tensor decompositions have been developed. In this way, CP decomposition can be done by selecting randomly elements from the original tensor (Battaglino et al., 2018), or using randomization to solve the problem on one or several smaller tensors before projecting back the result to the original space (Erichson et al., 2017; Sidiropoulos et al., 2014; Vervliet et al., 2014). Wang et al. (2015) proposed a fast yet provable randomized CP decomposition using FFT to perform tensor contraction. Randomization approaches have also be explored for fast approximation of other tensor decompositions, e.g., Tucker decomposition via sketching (Tsourakakis, 2010; Zhou et al., 2014) and tensor ring using tensor random projections (Yuan et al., 2019).

These methods are orthogonal to our approach and can be combined with it. As opposed to the aforementioned works, which focus on compression or efficiency, we explore tensor factorization methods in the context of adversarial defense, proposing a novel approach in which the weight tensor of each convolutional layer are parametrized using a randomized tensor decomposition that significantly hardens the model, increasing its robustness to a wide range of adversarial attacks. Our method is generic and can be applied to both real-valued networks and binary ones. In addition, it is orthogonal to the existing defence methods and can be combined with existing defenses such as adversarial training. The method is introduced in details in section 3.

## 3 DEFENSIVE TENSORIZATION

In this section, we introduce our method for the defense against adversarial attack. Our defense leverages a randomized higher-order factorization method, which is used as the basis for our defense. Typically, defensive methods relying on randomization do so by introducing sparsity in either the weights or the input activation tensors of the layers of the deep neural neural network. For instance, Dhillon et al. (2018) sparsify the input tensor, by stochastically pruning some of the activations and scaling up the remaining ones. Wang et al. (2018) apply sparsification to the weights directly using dropout both during training and testing. However, all these approaches degrade per-

formance by setting activations to zero, and, while rescaling the non-zero entries can mitigate the issue, increasing the sparsity (and therefore the efficiency of the defense) translates into large losses in performance. By contrast, we propose to rely on a latent parametrization of the layers using tensor decomposition. Intuitively, a latent subspace spanning the weights is learnt, along with a non-linear projection to and from that subspace. The sparsity inducing randomization is applied in the latent space. Upon projection, the resulting weights are dense and yet preserve the robustness against adversarial attacks. This allows us to built models that are both more robust to adversarial methods than existing works, while preserving high classification accuracy.

**Notation:** Throughout the papers, we denote vectors ($1^{\text{st}}$order tensors) as small bold letters $\mathbf{v}$, matrices ($2^{\text{nd}}$order tensors) as bold capital letters $\mathbf{M}$ and tensors, which generalize the concept of matrices for orders (number of dimensions) higher than 2, in capital calligraphic letters $\mathcal{X}$. The *n–mode* product is defined, for a given tensor $\mathcal{X} \in \mathbb{R}^{D_0 \times D_1 \times \cdots \times D_N}$ and a matrix $\mathbf{M} \in \mathbb{R}^{R \times D_n}$, as the tensor $\mathcal{T} = \mathcal{X} \times_n \mathbf{M} \in \mathbb{R}^{D_0 \times \cdots \times D_{n-1} \times R \times D_{n+1} \times \cdots \times D_N}$, with: $\mathcal{T}_{i_0,i_1,\cdots,i_n} = \sum_{k=0}^{D_n} \mathbf{M}_{i_n,k} \mathcal{X}_{i_0,i_1,\cdots,i_n}$.

**Latent high-order parametrization of the network:** We introduce tensor factorization in the context of deep neural networks. Note that this method is independent of the dimensionality of the input but we introduce it here, without loss of generality, for the case of a 4 dimensional kernel of 2D convolutions. Specifically, we consider a deep neural network composed of $L$ layers convolutional layers, interlaced with non-linearities $\Phi_l$, $l \in [1 .. L]$. Let's consider a convolutional layer $l \in [1 .. L]$, taking as input an activation tensor $\mathcal{X}_l$ and parametrized by a weight tensor $\mathcal{W}_l \in \mathbb{R}^{F,C,H,W}$, where $F, C, H, W$ correspond respectively to number of Filters (e.g. output channels), input Channels, Height and Width of the convolutional kernel. The output of that layer, after applying non-linearity, will be $\Phi(\mathcal{X}_l \star \mathcal{W}_l)$.

We introduce a latent parametrization of the weight kernel $\mathcal{W}_l$ by expressing it as a low-rank tensor, in this paper using a Tucker decomposition Kolda & Bader (2009). In other words we express $\mathcal{W}_l$ in a latent subspace as a core tensor $\mathcal{G}_l$. The mapping to and from this subspace is done via factor matrices $\mathbf{U}_l^F, \mathbf{U}_l^C, \mathbf{U}_l^H$ and $\mathbf{U}_l^W$: $\mathcal{W}_l = \mathcal{G}_l \times_0 \mathbf{U}_l^F \times_1 \mathbf{U}_l^C \times_2 \mathbf{U}_l^H \times_3 \mathbf{U}_l^W$.

**Randomizing in the latent subspace:** In addition to the above deterministic decomposition, we introduce, for each layer, a randomization term stochastically controlling the rank of the decomposition. To do so, we introduce diagonal sketching matrices $\mathbf{M}_F, \mathbf{M}_C, \mathbf{M}_H$ and $\mathbf{M}_W$, the diagonal entries of which are i.i.d. and follow a Bernoulli distribution parametrized by probability $\theta \in [0, 1]$. Specifically, we samples random vectors $\boldsymbol{\lambda}^F \in \mathbb{R}^O, \boldsymbol{\lambda}^C \in \mathbb{R}^C, \boldsymbol{\lambda}^H \in \mathbb{R}^H$ and $\boldsymbol{\lambda}^W \in \mathbb{R}^W$, the entries of which are i.i.d. following a Bernoulli distribution parametrized by probability $\theta$. We can then define the sketching matrices as $\mathbf{M}_O = \text{diag}(\boldsymbol{\lambda}_F), \mathbf{M}_C = \text{diag}(\boldsymbol{\lambda}_C), \mathbf{M}_H = \text{diag}(\boldsymbol{\lambda}_H)$ and $\mathbf{M}_W = \text{diag}(\boldsymbol{\lambda}_W)$.

This randomization is then applied not directly to the weight tensor $\mathcal{W}$, but rather in the low-rank subspace, effectively randomizing the *rank* of the convolutional kernel:

$$\tilde{\mathcal{W}}_l = \underbrace{(\mathcal{G}_l \times_0 \mathbf{M}_F \times \cdots \times_3 \mathbf{M}_W)}_{\text{randomized core } \hat{\mathcal{G}}_l} \times_0 \underbrace{\left(\mathbf{U}_l^F \mathbf{M}_F^\top\right) \times \cdots \times_3 \left(\mathbf{U}_l^W \mathbf{M}_W^\top\right)}_{\text{randomized factors } \hat{\mathbf{U}}_l^F, \hat{\mathbf{U}}_l^I, \hat{\mathbf{U}}_l^H, \hat{\mathbf{U}}_l^W} \quad (3)$$

This stochastic reduction of the rank can be done without affecting performance thanks to the over-parametrization of deep networks, which, while crucial for learning (Du & Lee, 2018; Soltanolkotabi et al., 2018), create large amounts of redundancies. In addition, since $\mathcal{G}_l \times_0 \mathbf{M}_F \times_0 \left(\mathbf{U}_l^F \mathbf{M}_F^\top\right) = \mathcal{G}_l \times_0 \left(\mathbf{U}_l^F \mathbf{M}_F^\top \mathbf{M}_F\right)$, and $\mathbf{M}_F, \mathbf{M}_I, \mathbf{M}_H$ and $\mathbf{M}_W$ are idempotent, eq. 3 can be simplified to:

$$\tilde{\mathcal{W}}_l = (\mathcal{G}_l \times_0 \mathbf{M}_F \times \cdots \times_3 \mathbf{M}_W) \times_0 \mathbf{U}_l^F \times \cdots \times_3 \mathbf{U}_l^W = \tilde{\mathcal{G}}_l \times_0 \mathbf{U}_l^F \times \cdots \times_3 \mathbf{U}_l^W \quad (4)$$

In other words, we sketch the core tensor, then project it back using the original factors. An important detail is that the randomization terms from the above equation, $\tilde{\mathcal{G}} = \mathcal{G} \times_0 \mathbf{M}_F \times \cdots \times_N \mathbf{M}_W$ is never explicitly computed using actual tensor contractions. Instead, the elements sampled are directly selected from the core and the corresponding factors, which is much more computationally effective. For the binary case, we additionally apply the process from eq. (5) by plugging in eq. (4).

The randomization being done in the latent subspace, it induces no sparsity, unlike pruning or dropout based methods and the reconstructed weights are dense. Since the weights are learnt end-to-end with randomization on the latent cores, the network naturally learns not to rely on any single latent component for prediction, thus learning intrinsically more robust representations. The result is a network that is naturally more robust to perturbations in the inputs.

# 4 EXPERIMENTAL SETTING

In this section, we detail the experimental setting used for the experiments for image-based classification and audio-based classification.

## 4.1 DATASETS

We conducted experiments on two widely used databases for image-based classification and audio-based classification, respectively:

**CIFAR-10 (Krizhevsky & Hinton, 2009)** is a widely used image classification dataset consisting of $60,000$ images of size $32 \times 32$px in 10 classes, equally represented. The dataset is divided into $50,000$ images for training and $10,000$ for testing. We did not use any particular data augmentation, beside random horizontal flipping.

**Speech Command (Warden, 2018)** is an audio recognition dataset comprised of $105,000$ 1-second utterances of words from a large number of users spanning over a small vocabulary. The objective is to recognize among ten spoken words: *yes, no, up, down, left, right, on, off, stop, go*, in addition to recognizing words outside the vocabulary as *unknown*, and detecting *silence*. The dataset is balanced and all audio recordings are captured with a sampling frequency of 16 KHz. We use a 80%-10%-10% splits for training, validation and testing respectively.

## 4.2 IMPLEMENTATION DETAILS

**Training the model**  All our CIFAR-10 (Krizhevsky & Hinton, 2009) experiments were conducted using a ResNet-18 (He et al., 2016) architecture. The network was trained for $350$ epochs using SGD with momentum ($0.9$) and a starting learning rate of $0.1$ that was dropped at epoch $150$ and $250$ by a factor of $0.1$. The weight decay was set to $1e - 6$. In order to accelerate the training process the models with $\theta < 1$ were initialized from a pretrained model that was trained without stochasticity ($\theta = 1$). The binary counterpart models were trained following the method proposed by Rastegari et al. (2016) using the same optimizer and learning scheduler as for the real-valued ones. For the experiments conducted on the Speech Command dataset we build on the SoundNet5 (Aytar et al., 2016) architecture containing 5 convolutional layers *[in_channels, out_channels, kernel, stride, padding]*: $[1, 16, (1 \times 64), (1 \times 2), (0 \times 32)], [16, 32, (1 \times 32), (1 \times 2), (0 \times 16)], [32, 64, (1 \times 16), (1 \times 2), (0 \times 8)], [64, 128, (1 \times 8), (1 \times 2), (0 \times 4)], [128, 256, (1 \times 4), (1 \times 2), (0 \times 2)]$ and 2 linear ones: $[512, 256]$ and $[256, 12]$. Each convolutional layer was followed by a max-pooling operation. We trained all of the audio models using Adam (Kingma & Ba, 2014) for 50 epochs with an initial learning rate set to $0.01$ that was dropped by $0.1\times$ at epoch 25 and 35. The binarization and re-parametrization follows the same procedure as for CIFAR-10.

**Attacking the model**  For FGSM, we run the attack for various values of $\epsilon = \{1, 2, 4, 8, 16, 32, 64, 128\}$ (for an image range $[0 .. 255]$) across the entire validation/testing set averaging the results over 10 runs. On the Speech Command dataset, since the raw data is in the range $[-1, 1]$, we scaled the value of $\epsilon$ accordingly, running it for the following values $\epsilon = \{0.008, 0.032, 0.063\}$. For the iterative methods BIM and PGD, we follow Kurakin et al. (2016) and Song et al. (2017) setting the step size to 1 and the number of iterations to $\lfloor \min(\epsilon + 4, 1.25\epsilon) \rfloor$.

**Threat Model:**  we assume that the attacker has access to everything (e.g. the architecture of the network, its

| Attack | | Method | | | |
|---|---|---|---|---|---|
| | | | | **Ours** | |
| **Type** | $\epsilon$ | **Baseline** | $\theta = 0.95$ | $\theta = 0.9$ | $\theta = 0.8$ |
| **Clean** *(no attack)* | | **95.3** | 94.5 | 93.0 | 90.1 |
| FGSM | 2 | 48.7 | 84.9 | **86.8** | 83.4 |
| | 8 | 22.5 | 65.4 | 69.9 | **71.5** |
| | 16 | 12.7 | 54.0 | 56.0 | **60.3** |
| BIM | 2 | 23.0 | 60.2 | 69.5 | **71.8** |
| | 8 | 0.0 | 26.6 | 33.1 | **45.5** |
| | 16 | 0.0 | 27.0 | 33.0 | **42.4** |
| PGD | 2 | 22.9 | 64.4 | 72.3 | **76.2** |
| | 8 | 0.0 | 27.0 | 28.1 | **42.9** |
| | 16 | 0.0 | 22.4 | 27.4 | **34.3** |

Table 1: **Real-valued network performance on CIFAR-10** for FGSM, BIM and PGD attacks with $\epsilon \in \{2, 8, 16\}$.

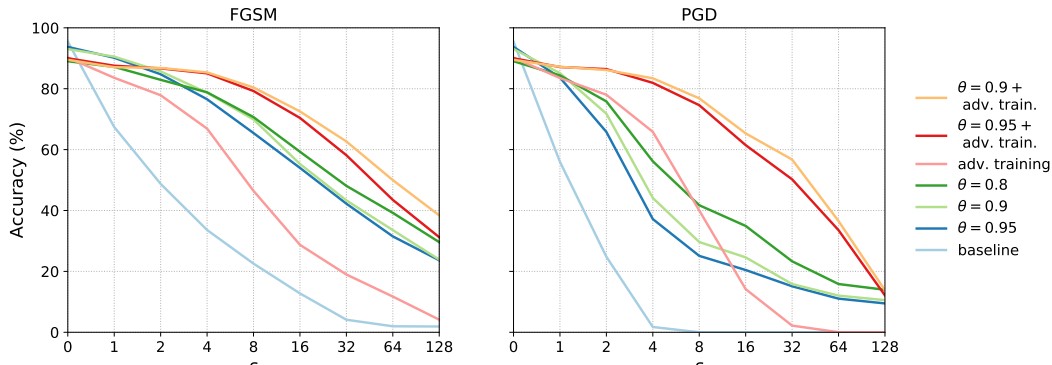

Figure 1: FGSM (left) and PGD (right) attacks on the CIFAR-10 image classification dataset for various values of $\epsilon$ with and without adversarial training. Notice that our method alone surpasses the strong adversarial training defence. When combined together the robustness is increased even further. Results are averaged over 10 runs.

weights, inputs, outputs, training process and gradients, etc) except the random seed used for sampling the Bernoulli random variables.

All of our models were implemented using PyTorch (Paszke et al., 2017) and trained on a single Titan X GPU. The latent, randomized tensor factorization was implemented using TensorLy (Kossaifi et al., 2019b). For the adversarial attacks we used the FoolBox (Rauber et al., 2017) package.

## 5  RESULTS

In this section, we empirically demonstrate the robustness property of our proposed method against adversarial examples by extensively evaluating it on CIFAR-10 and comparing it against existing state-of-the-art defense techniques. Moreover, we show that our approach can be combined with adversarial training based techniques leading to further robustness gains. All the experiments are run using the experimental setup described in Section 4.

**Robustness to adversarial attacks:**  When evaluated using the FGSM attack on the CIFAR-10 dataset our method is significantly more robust than the baseline approach, especially for high values of $\epsilon$ (see Table 1 and Fig. 1). Furthermore, the results presented in Table 1 show that for lower values of $\theta = 0.8$, our network significantly decreases the typical high attack success-rate achieved by strong iterative attack algorithms such as BIM and PGD.

**Comparison to the State-of-the-Art.**  To better understand the performance of our method, we compare it against existing state-of-the-art defense techniques such as: DQ (Lin et al., 2019) in which the authors attempt to reduce the propagation of the adversarial attacks inside the network by controlling the Lipschitz constant, features squeezing (Xu et al., 2017) that simple reduces the dimensionality of the search space by controlling the color bit depth of each pixel while applying spatial smoothing and finally, adversarial training in two variants: R+FGSM as proposed in (Lin et al., 2019) and PGD (Madry et al., 2017). Where the later (i.e. adversarial training) is considered to be one of the strongest defence techniques developed. For feature-squeezing we used 5 bits for image color reduction combined with a $2 \times 2$ median filter. For adversarial training as in (Madry et al., 2017; Kurakin et al., 2016) we sample the number of steps (for PGD) and $\epsilon$ randomly. As the results from Table 2 show, our method consistently outperforms existing defense strategies for various attacks (FGSM, BIM, PGD), across different values of $\epsilon = \{2, 8, 16\}$. Furthermore, when combined with adversarial training our method can further increase its resilience to attacks.

**Defending against omniscient attackers**  Another interesting question is whether our defense strategy would still work against an *omniscient* attacker, i.e., an attacker with access to the full un-randomized weights. We trained a network for $\theta = 0.9$. Then, during training, we first generate an adversarial example using the full, un-randomized weights (i.e. $\theta = 1$) and test it using the stochasticity (i.e. $\theta = 0.9$). As can be seen in Table 3, our network is still robust against these

| Method | clean | FGSM $\epsilon = 2/8/16$ | PGD $\epsilon = 2/8/16$ |
|---|---|---|---|
| Normal | 95.4 | 48.7/22.6/12.9 | 22.9/0/0 |
| DQ (Lin et al., 2019) | **95.9** | 68/53/42 | 62/4/0 |
| Feature squeezing (Xu et al., 2017) | 94.1 | 61/35/27 | 64/2/0 |
| Adv. training+R FGSM (Lin et al., 2019) | 91.6 | 81/52/38 | 84/43/11 |
| Adv. training PGD (Madry et al., 2017) | 86.6 | 74/46/31 | 76/44/20 |
| **Ours ($\theta = 0.8$)** | 90.1 | 83.4/71.5/60.3 | 76.2/42.9/34.3 |
| **Ours ($\theta = 0.95$) + Adv. training PGD** | 89.5 | **86.5/81.4/70.1** | **84.9/75.4/59.8** |

Table 2: **Comparison against various defense methods against FGSM and PGD** with $\epsilon = \{2, 8, 16\}$ on the CIFAR10 dataset. Notice that our method, especially when combined with adversarial training, significantly outperforms other state-of-the-art methods.

attacks, despite the attacker having full access to the weights. Similar behaviour can be observed for other values of $\theta$. Note that this is an extreme scenario and in general, the weights could be stored separately (and safely) with the network getting, at each time, the randomly reconstructed weights.

## 6 ABLATION STUDIES

To further validate our findings we test out approach on two different scenarios: on fully binarized networks (Section 6.1) and audio classification task (Section 6.2).

| $\epsilon$ | FGSM | PGD | BIM |
|---|---|---|---|
| **2** | 91.1 | 82.5 | 81.9 |
| **8** | 85.4 | 74.1 | 80.4 |
| **16** | 82.4 | 53.9 | 80.1 |

### 6.1 ATTACKING BINARY NEURAL NETWORKS

With the growing popularity of deep learning based methods and the constant need of incorporating such approaches on mobile devices, network quantization has emerged as a potential hardware-friendly solution, which squeezes the original network by reducing the number of bits required to represent the model parameters. In its most extreme form, binarization, the weights and features are represented using a single bit (Courbariaux et al., 2016; Rastegari et al., 2016; Bulat & Tzimiropoulos,

Table 3: **Robustness against attacker with access to the un-randomized weights**. FGSM, BIM and PGD attacks with $\epsilon \in \{2, 8, 16\}$ are computed using the full (un-randomized) weights (i.e. $\theta = 1$) and used against the **same** network with the **same** weights but with $\theta = 0.9$.

2017). The typical approach quantizes the network using the *sign* function (Courbariaux et al., 2015), however this introduces high quantization errors that hinder the learning process. To alleviate this, a real-valued scaling factor is introduced by Rastegari et al. (2016). In this work we binarize the network following Rastegari et al. (2016):

$$\mathcal{I} * \mathcal{W} = (\text{sgn}(\mathcal{I}) \circledast \text{sgn}(\mathcal{W})) \odot \mathcal{K}\alpha, \tag{5}$$

where $\mathcal{I} \in \mathbb{R}^{c \times w_{in} \times h_{in}}$ and $\mathcal{W} \in \mathbb{R}^{c \times h \times w}$ denote the input and respectively the weight of the $L$-th convolutional layer, $\alpha \in \mathbb{R}^{c \times 1 \times 1}$ represent the weight scaling factor and $\mathcal{K} \in \mathbb{R}^{1 \times h_{out} \times w_{out}}$ the input scaling factor. Both $\alpha$ and $\mathcal{K}$ are computed analytically as proposed by Rastegari et al. (2016).

While it was previously thought that such binarized networks are more resilient to adversarial attacks (Khalil et al., 2018; Liu et al., 2018; Galloway et al., 2017) than their real-valued counterpart, in this work we confirm the recent findings of (Lin et al., 2019) by showing that in fact it is the opposite, i.e., binary networks are more susceptible to adversarial attacks. Typically, during the training phase the derivative of the quantization function (sgn) is approximated using a STE (e.g., an identity function clipped to $[-1, 1]$ in this case). The same estimator can be used during the attacking phase and it often leads to a high rate of success of the attacks. Interestingly, the results in Table 4 show that we can go one step further by approximating the derivative of the *sgn* function using $\tanh(x)$ and $\tanh(0.75 * x)$ respectively. The use of these approximations make the binary networks become more sensitive to the attacks. When evaluated on a binary network on the CIFAR-10 dataset, in accordance with the behaviour found on real-valued models, our method shows significant improvements across the entire range of values and attacks tested (see Table 4 and Fig. 1). We note that for the binary case, since such networks have a lower representational capacity, we set $\theta = 0.99$.

| Attack | | Method | | | |
|---|---|---|---|---|---|
| **Type** | $\epsilon$ | **Baseline** | | | **Ours** ($\theta = 0.99$) |
| | | Identity | $\tanh(x)$ | $\tanh(0.75x)$ | |
| **Clean** *(no attack)* | | 83.7 | 83.7 | 83.7 | 80.0 |
| **FGSM** | **2** | 36.6 | 34.5 | 34.1 | **76.9** |
| | **8** | 6.9 | 6.1 | 5.8 | **65.0** |
| | **16** | 4.3 | 3.4 | 3.0 | **58.7** |
| **BIM** | **2** | 37.0 | 34.7 | 35.1 | **66.3** |
| | **8** | 0.0 | 0.0 | 0.0 | **46.4** |
| | **16** | 0.0 | 0.0 | 0.0 | **44.0** |
| **PGD** | **2** | 41.7 | 38.7 | 39.8 | **67.5** |
| | **8** | 0.1 | 0.0 | 0.0 | **47.9** |
| | **16** | 0.0 | 0.0 | 0.0 | **41.5** |

Table 4: **Binary network performance on CIFAR-10** for FGSM, BIM and PGD attacks with $\epsilon \in \{2, 8, 16\}$. Our approach is significantly more robust, especially against iterative attacks.

## 6.2 DEFENSIVE TENSORIZATION FOR AUDIO CLASSIFICATION

To further demonstrate the generalizability of our approach, we next consider adversarial attacks on the audio domain. In-line with the latest success in audio recognition, we consider an end-to-end audio model following SoundNet (Aytar et al., 2016) architecture, that operates directly on the raw audio signal, without requiring any feature extractions (e.g., MFCC or log mel-spectrogram). We found that the end-to-end models show higher degree of vulnerability to the adversarial attacks, e.g., around 6% absolute degradation compared to the model operating on log mel-spectrogram. In case of the small vocabulary audio recognition task, we only consider FGSM attack and summarize our findings in Table 5. With a higher degree of stochasticity (i.e. $\theta = 0.9$), both the real and the binarized model exhibit much higher resilience to the adversarial attacks.

| Quantization | $\epsilon$ | **Baseline** | **Defensive Tensorization** | | |
|---|---|---|---|---|---|
| | | | $\theta = 0.99$ | $\theta = 0.95$ | $\theta = 0.9$ |
| **Real** | No attack | **93.8** | 92.0 | 89.6 | 88.1 |
| | **0.008** | 33.0 | 49.6 | 58.2 | **61.0** |
| | **0.032** | 14.9 | 33.0 | 40.2 | **44.2** |
| | **0.063** | 7.6 | 23.8 | 31.8 | **35.7** |
| **Binary** | No attack | 88.0 | **89.0** | 83.5 | 83.2 |
| | **0.008** | 12.2 | 50.1 | 54.4 | **56.0** |
| | **0.032** | 3.0 | 31.5 | 35.6 | **40.2** |
| | **0.063** | 0.2 | 26.7 | 30.3 | **30.5** |

Table 5: **Performance on Speech Command** for FGSM attacks with $\epsilon \in \{0.008, 0.032, 0.063\}$ using both binary and real-valued models. Notice that our approach is significantly more robust.

## 7 CONCLUSION

In this paper, we propose defensive tensorization, a novel adversarial defence technique that leverages a latent high order factorization of the network. Randomization is applied in the latent subspace, therefore resulting in dense reconstructed weights, without the sparsity or perturbations typically induced by the randomization. We empirically demonstrate that our approach makes the network significantly more robust to adversarial attacks. Contrarily to a widely spread belief, we observe that binary networks are *more* sensitive to adversarial attacks than their real-valued counter-part. We show that our method significantly improves robustness in the face of adversarial attacks for both binary and real-valued networks. We demonstrate this empirically through thorough experimentation on image and audio classification.

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

## A APPENDIX

**Algorithm for defensive tensorization** We summarize in Algorithm. 1 the steps for inference using our proposed defensive tensorization.

---

**Algorithm 1** Defensive tensorization

---

1: **Input** sample $\mathcal{X}_0$.
2: **for each** layer $l$ **do**
3:     **Sample** random vector $\boldsymbol{\lambda}_F, \cdots, \boldsymbol{\lambda}_W \sim$ Bernouilli $(\theta)$
4:     **Create** sketch matrices $\mathbf{M}_F = \text{diag}(\boldsymbol{\lambda}_F), \cdots, \mathbf{M}_W = \text{diag}(\boldsymbol{\lambda}_W)$
5:     **Sketch** the latent Core $\tilde{\mathcal{G}}_l = \mathcal{G}_l \times_0 \mathbf{M}_F \times \cdots \times_3 \mathbf{M}_W$
6:     **Reconstruct** $\mathcal{W}_k$ using Equation (3)       ▷ Randomized *dense* reconstruction
7:     $\mathcal{X}_{l+1} \leftarrow \Phi_l(\mathcal{W}_l \star \mathcal{X}_l)$     ▷ Convolve $\mathcal{X}_l$ with the randomized reconstruction and apply $\Phi_l$
8: **return** $\mathcal{X}_L$

---

**Low-dimensional visualization of loss landscape:** To visually assess the impact of both our randomization scheme and adversarial training on the optimization landscape, we visualise the evaluation of the loss in a fixed neighbourhood around an unseen data point. Specifically, given a model and a data point $x$, we visualize the loss function learned by the model in the close proximity of $x$. For clarity, we visualise the loss in a 2–dimensional space, by selecting the two relevant axis: the direction of the gradient at $x$ (x-axis) and a randomly chosen direction, orthogonal to the direction of gradient (y-axis). Next, a mesh-grid is constructed by sampling uniformly points along these two directions for the range $[-0.5, 0.5]$. Then a contour plot is constructed by evaluating the losses for all points on the mesh-grid. Examples of loss-surfaces are shown in Fig. 2, 3 and 4.

**Effect on the optimization landscape:** Intuitively, the randomization (which is done in the *latent* subspace of the decomposition, not on the weights themselves), changes the loss function, at each pass, making it hard to converge to a fixed attack due to the presence of many spurious minimums. This can be seen by looking at the landscape of the loss function around an arbitrary sample in Fig. 2, 3. For the adversarial case, the landscape is inline with the finding of Madry et al. (2017), where the authors show the adversarial training smooths the loss space around 0. This is even more noticeable for the method that combines our approach with adversarial training (see Fig. 4).

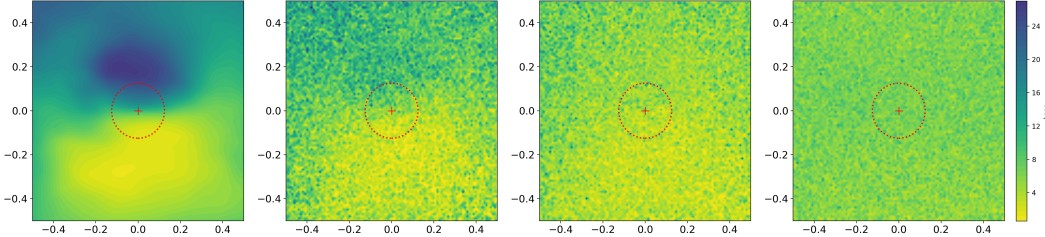

Figure 2: **Contour plot of the loss surface** of the real model evaluated on the $l_\infty$ neighbourhood of a CIFAR-10 image for $\theta = \{1, 0.95, 0.9, 0.8\}$ (from left to right). The direction of the gradient was computed for $\theta = 1$ and a random, orthogonal direction. These were kept fixed for the subsequent plots $\theta = 0.95, 0.9$ and $0.8$. The red circle denotes the $\epsilon = 32$ neighbourhood.

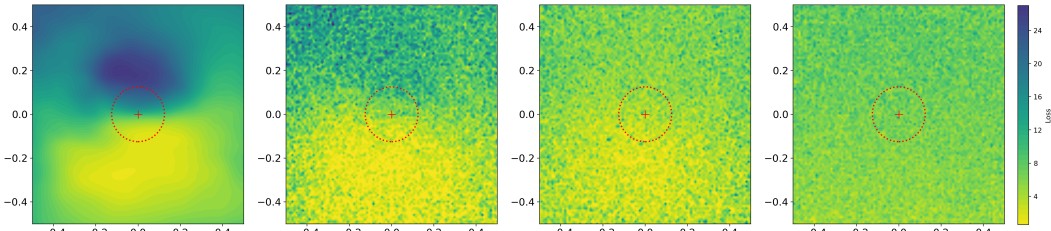

Figure 3: **Contour plot of the loss surface** of the real model evaluated on the $l_\infty$ neighbourhood of a CIFAR-10 image for $\theta = \{1, 0.95, 0.9, 0.8\}$ (from left to right). For each plot (i.e. each value of $\theta$), we recomputed the direction of the gradient. The red circle denotes the $\epsilon = 32$ neighbourhood.

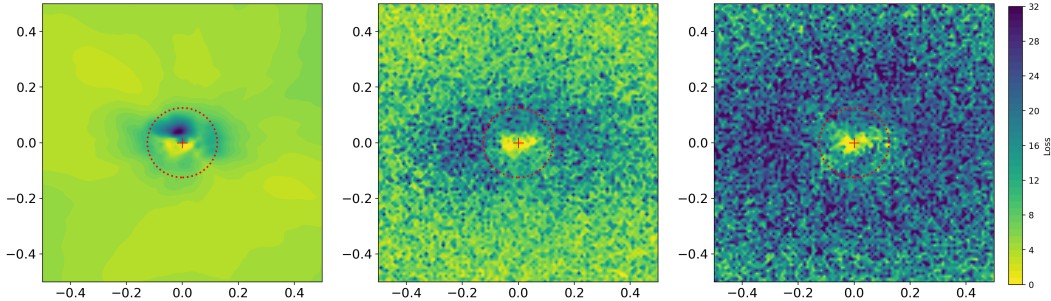

Figure 4: **Contour plot of the loss surface** of our model, adversarially trained model for various values of $\theta$ evaluated on the $l_\infty$ neighbourhood of an unseen CIFAR-10 image. The same direction was used for all three plots. The red circle denotes the $\epsilon = 32$ neighbourhood.

