# OpenReview forum: "Defensive Tensorization: Randomized Tensor Parametrization for Robust Neural Networks"
_ICLR.cc/2020/Conference — Reject_

### Official Review · AnonReviewer3 · 2019-10-23
**Official Blind Review #3**

**Rating:** 6

**Review:**

The authors propose  to use randomized tensor factorization in the weight space as a defense to adversarial attack, which builds upon the existing works on using randomization on the weights or activation as a defense methods.

Pros:
1. The idea of using randomized tensor factorization for dense is novel
2. It seems that this defense is robust to large perturbation (epsilon), and the accuracy on clean data is high when combined with PGD adv.training.

Cons:
1.  I don't understand why using randomization in the latent space of the weights can retain the classification accuracy on clean data. The authors say that this is because both the weights and the activations are not sparse. But I don't understand the relation between sparsity and accuracy. Can the author can provide some evidence on this (probably from the previous acceleration literatures). Besides, I think an accuracy of 90.1 on CIFAR10 (Tab.2) is not high.
2. As in the review written by Anthony Wittmer, the author should include experiments to check the obfuscated gradient issue.

**Experience Assessment:**

I have read many papers in this area.

**Review Assessment: Checking Correctness Of Derivations And Theory:**

I assessed the sensibility of the derivations and theory.

**Review Assessment: Checking Correctness Of Experiments:**

I did not assess the experiments.

**Review Assessment: Thoroughness In Paper Reading:**

I read the paper at least twice and used my best judgement in assessing the paper.

---

> ### Author Response · Authors · 2019-11-14
> **Response to Reviewer 3**
>
>
> $\bullet \textit{ 1. I don't understand why using randomization in the latent space of the weights can retain the classification accuracy on clean data. }$
>
> Using randomization in the latent space has several advantages. First, the distributions of the reconstructed weights is preserved, despite the randomness, since the regularization during training happens is in the latent subspace, not directly on the weights. In particular, no sparsity is induced on the weights. On the other hand, existing randomizations methods that *do* induce sparsity, such as Stochastic Activation Pruning or Dropout based approaches require correcting the magnitudes after pruning. In addition, these approaches are significantly less robust than our approach to adversarial attacks.  In addition, when training deep convolutional neural networks, there is an evidence that over-parameterization is crucial (Du & Lee, 2018; Soltanolkotabiet al., 2018). Our latent parameterization allows for both over-parameterization in the reconstruction space and preserving performance while still having randomization in the latent space, which can be controlled with a large values of $\theta$.
>
> $\bullet \textit{ Besides, I think an accuracy of 90.1 on CIFAR10 (Tab.2) is not high. }$
>
> Regarding performance (90.1), there is a tradeoff between accuracy on the clean set and robustness to adversarial attacks. Please note that adversarial training alone (the go-to method) trained in the same exact setting, has a performance of only 86.6 on the clean set, which is significantly worse than ours. In addition, our approach is more robust to adversarial attacks in all cases.
>
> $\bullet \textit{ 2. As in the review written by Anthony Wittmer, the author should include experiments to check the obfuscated gradient issue.}$
>
> Thank you for your comment, we have ran this additional experiment. As mentioned to Anthony Wittmer, in Section 5, "Defending against omniscient attacker", we hoped to address that very point. In that scenario, the attacker has access to the full (unrandomized) weights and uses these to perform the attack. The idea is that these unrandomized weights could be obtained by accumulating forward passes as suggested in BPDA[1].
> Please note that an important point of our approach is that we do not randomize the weights directly. Instead, we apply randomization in the latent subspace spanned by the low-rank structure imposed. You can think of it as a stochastic regularization applied to the *rank* of the tensor factorization.
>
> One could argue that applying [1] would result in different results (e.g. [1] acts as an ensemble of models). To verify, in addition to the above scenario and following your comments, we ran the following additional experiment (BPDA [1]):
> at each iteration of gradient descent, for each convolutional layer, instead of taking a step in the direction of $\nabla_x f(x)$  we move in the direction of $\sum_{i=1}^{k}\nabla_x f(x)$ where each pass has the weights randomized in the latent space using our approach.
>
> We report here the accuracy (on CIFAR 10), obtained using our best model, for various values of $\epsilon=2,8,16$:
>
> $\begin{array} {|r|r|}\hline Method & 2 & 8 & 16 \\ \hline BPDA & 83.3 & 54.9 & 43.8 \\ \hline  \end{array}$
>
> While in [1] the authors use $k=10$ we try with up to $k=20$ but without noticing any significant increase in the success rate of the attack. The PGD attack itself was run for 500 iterations as in [1]. These results are in line with the results we reported in the paper, see Table 2 in the manuscript.

---

### Official Review · AnonReviewer1 · 2019-10-24
**Official Blind Review #1**

**Rating:** 6

**Review:**

This paper tackles the problem of designing neural network architectures that are robust to adversarial attacks. Several defense techniques against adversarial attacks have been proposed, mainly adversarial training (train on perturbed inputs) and introducing random perturbation to the weights or activations of the network. The paper claims that one limitation of the second approach is that it introduces artifacts (e.g. sparsity). The authors propose a simple but original idea to address this issue: parameterize the network's weight matrices as low rank tensors (in the Tucker format) and randomize the weights by sketching the core tensor of the Tucker decomposition (in effect, the sketching amounts to randomly setting fibers of the core tensor to 0).

I think this paper can be relevant to the community but I am not confident that this is an important contribution. The idea is interesting and addresses the problem of sparsity artifacts in randomized defense strategies, but it does not appear clearly why using tensor decomposition techniques is a sound approach for designing robust networks (besides overcoming sparsity artifacts). I believe there may be more fundamental (theoretical, principled) arguments to motivate the approach, but this is not explored in the paper: the idea is interesting but not supported by much theoretical insight. Yes, using Tucker decomposition allows one to have randomized but still dense weights. Is it the only reason to use tensor decomposition? Why not do the same with a simple low rank matrix for example?

The experimental section is developed but I find the experimental setting not clearly described (e.g., what is the metric? is it the accuracy over adversarial examples?). Maybe this is because I am not familiar with the adversarial defense literature.

In conclusion, I am a bit on the fence for this paper. The idea is interesting and definitely worth exploring but to me a more thorough discussion and analysis of why tensor decomposition techniques are relevant is missing. Still, the approach is original and this paper may spark future work further exploring these questions, so I recommend acceptance.

* Comments / Questions *

- Paragraph "Latent high-order parametrization of the network". If I understand correctly, the core tensor G in the decomposition as the same size of W, so at this stage W is not parameterized as a low rank tensor (W is actually over-parmeterized). This is only when the stochastic vectors \lambda are introduced that the Tucker rank of W is implicitly reduced. This could be clarified.

- Is stochasticity preserved at test time (unlike when using dropout but like in Wang, et al. (2018))?

- What is the metric used in Table 1 to compare the models?

- Would it make sense to explore other tensor decomposition models (e.g. CP, tensor train, tensor ring, ...)? Are there any particular reasoning motivating the choice of Tucker?




**Experience Assessment:**

I do not know much about this area.

**Review Assessment: Checking Correctness Of Derivations And Theory:**

I carefully checked the derivations and theory.

**Review Assessment: Checking Correctness Of Experiments:**

I did not assess the experiments.

**Review Assessment: Thoroughness In Paper Reading:**

I read the paper at least twice and used my best judgement in assessing the paper.

---

> ### Author Response · Authors · 2019-11-14
> **Response to Reviewer 1**
>
>
> $\bullet \textit{Tucker decomposition allows one to have randomized but still dense weights. Is it the only reason to use tensor decomposition?}$
> $\textit{Why not do the same with a simple low rank matrix for example?}$
>
> Thank you for your thoughtful feedback, these points will be clarified in the paper.
> i) Tensor methods have the ability to leverage the multi-linear structure in the data, weight and activations. This is a property that is desirable when building robust neural networks. It can be noted that the matrix case is a special case of our approach. Specifically, in Equation 3, this can be obtained by setting $M_I = M_H = M_W = U_l^{I} = U_l^{H}  = U_l^{W} = \mathbf{I} $. The equality can then be rewritten in term of the mode-1 unfolding to obtain the matrix case. We will add this and a discussion in the final manuscript. In addition, to validate the above hypothesis, we ran an additional experiment on CIFAR-10, using a ResNet-18 architecture. We run both the matrix and tensor version of our method, for the same value theta,. Notice that the tensor decompositions offers consistent gains over the matrix ones. The results are reported in the table below:
>
> $\begin{array} {|r|r|}\hline Method & Clean & FGSM(2/8/16) & BIM(2/8/16) & PGD(2/8/16) \\ \hline Ours & 94.5 & 84.9/65.4/54.0 & 60.2/26.6/27.0 & 64.4/27.0/22.4 \\ \hline matrix~decomp. & 93.7 & 76.9/52.8/40.9 & 44.6/17.5/18.2 & 50.0/16.9/15.0 \\ \hline  \end{array}$
>
> ii) The regularization during training happens is in the latent subspace, and the network is learnt end-to-end with this regularization, and thus preserving the distributions of the reconstructed weights, despite the randomness. In particular, no sparsity is induced on the weights, as opposed to existing methods such as Stochastic Activation Pruning or Dropout based approaches.
>
> iii) Lastly, when training deep convolutional neural networks, there is an evidence that over-parameterization is crucial  (Du & Lee, 2018; Soltanolkotabiet al., 2018) Our latent parameterization allows for both over-parameterization in the reconstruction space and preserving performance while still having randomization in the latent space, which can be controlled with a large values of $\theta$.
>
> * Simon  S  Du  and  Jason  D  Lee.   On  the  power  of  over-parameterization  in  neural  networks  with quadratic activation. In ICML, 2018.
> * Mahdi Soltanolkotabi, Adel Javanmard, and Jason D Lee. Theoretical insights into the optimization landscape of over-parameterized shallow neural networks. IEEE Transactions on Information Theory, 2018.
>
> $\bullet \textit{The experimental section is developed but I find the experimental setting not clearly described (e.g., what is the metric? is it the accuracy over adversarial examples?).}$
>
> All the results reported throughout the paper are in terms of Top-1 accuracy computed across the entire test set and reflect the changes in accuracy due to the adversarial attacks as we increase the magnitude of the attacks. We will further clarify this in the paper.
>
> $\bullet \textit{The core tensor G in the decomposition as the same size of W, so at this stage W is not parameterized as a low rank tensor (W is actually over-parmeterized).}$
> $\textit{When the stochastic vectors \lambda are introduced the Tucker rank of W is implicitly reduced. This could be clarified. }$
>
> Yes, we use a full-rank decomposition, since, as pointed out by the reviewer, the focus is the randomization, not the low-rank structure. We will make this clear in the paper.
>
> $\bullet \textit{- Is stochasticity preserved at test time (unlike when using dropout but like in Wang, et al. (2018))?}$
>
> Yes, we preserve the stochasticity at test time (indicated by the value of theta in each table). We will make this clear in the paper.
>
>  $\bullet  \textit{- What is the metric used in Table 1 to compare the models? }$
> The metric used in Table 1 and through this paper is the Top-1 accuracy. Thanks for pointing it out, we will make sure to mention this in the caption and body of the manuscript.
>
> $\bullet \textit{- Would it make sense to explore other tensor decomposition models? Are there any particular reasoning motivating the choice of Tucker? }$
>
> It would be interesting to explore other decompositions. The randomization is general and does not depend on a specific decomposition. In this paper, we selected a Tucker structure as it is well suited to our use-case as it induces a latent subspace (represented by the core), with a multi-linear mapping to and from that subspace defined by the factors of the decomposition. The stochastic regularization is applied in the latent subspace whilfrom which the actual weights are then reconstructed using the multi-linear mapping.
>
> CP is a special case of Tucker where the core is super-diagonal so we can assume the performance would be similar. Other decompositions such as Tensor-Train would be an interesting experiment which we leave for future work due to time-constraint.

---

### Official Review · AnonReviewer2 · 2019-10-25
**Official Blind Review #2**

**Rating:** 6

**Review:**

In this paper, the authors propose a randomization-based tensorization framework towards robust network learning. The high-level idea of this work is to reparameterize the network parameters W of each layer with low-rank tensors, where the factor matrices are injected with randomization through randomly sampled sketching matrices. Since the randomization is is done within a subspace than directly on the weight matrix itself, the authors claim that this brings certain advantages such as less sparsity.

Strengths:
+ Well-written paper with good clarity and technical correctness.
+ Interesting idea with novelty.
+ Good ablation study with clear performance improvement from the proposed framework.
+ Good applications with binarized networks and audio classification.

Weaknesses:
- Insufficient and badly conducted comparative study with recent SOTAs.
- Insufficient experiment with larger datasets (such as CIFAR-100) or enough variety of datasets (such as SVHN).
- No direct experiment verification that supports the advantage of randomization in a subspace
- No discussions on the training complexities and the extendability to large-scale datasets/networks, such as ImageNet/ResNet-101.
- Missing citation and comparison to the following two SOTAs:
1. Xie et al., Feature Denoising for Improving Adversarial Robustness, CVPR19
2. Mustafa et al., Adversarial Defense by Restricting the Hidden Space of Deep Neural Networks, ICCV19

Comments:
I consider the idea of this paper novel and interesting. Considering tensor factorization with randomization for network robustness makes a lot of sense but overall the experiments of this paper are not well-conducted towards comparative studies with other SOTAs, although ablation study shows the considerably improved robustness from the proposed method. The main concerns of this paper lie in several aspects:
1. It seems that the authors did not report their comparison to recent SOTAs (such as Lin et al, 2019) comprehensively enough, nor were the benchmark measures (missing several other attacks, especially black box ones), datasets and backbones fully aligned. It is unclear how much the architecture of a backbone can impact the fairness of comparison. There is also no apples to apples comparison to directly verify the advantage of this work over non-subspace-based randomization method.
2. The authors failed to cite and compare to recent two SOTAs (listed above) which conduct large-scale experiments with bigger models. And there is no discussion about the extendability/generalizability of the proposed method to these data and models. Therefore, the contributions of this work somehow become less convincing.

Minor typos:
In page 4 "Randomizing in the latent subspace":
\lambda^F \in R^O --> \lambda_F \in R^F
M_O = diag(\lambda_F) --> M_F = diag(\lambda_F)
please unify subscripts/superscripts for all \lambda.

**Experience Assessment:**

I have read many papers in this area.

**Review Assessment: Checking Correctness Of Derivations And Theory:**

I carefully checked the derivations and theory.

**Review Assessment: Checking Correctness Of Experiments:**

I assessed the sensibility of the experiments.

**Review Assessment: Thoroughness In Paper Reading:**

I read the paper at least twice and used my best judgement in assessing the paper.

---

> ### Author Response · Authors · 2019-11-14
> **Response to Reviewer 2 [part 1/2]**
>
> $\bullet \textit{- Insufficient and badly conducted comparative study with recent SOTAs.}$
> In the experimental evaluation section of the paper we followed the same setup as previous landmarks and state-of-the-art papers in the field, such as (Lin et al, 2018, Xu et al, 2017, Madry et al, 2019). We also reported results with an additional architecture for audio data, and experimented with binary neural networks.
> Furthermore, based on your suggestion, we also ran additional experiments which we will add to the final version of the paper. Specifically, we ran experiments on CIFAR-100, compared with the method suggested by reviewer 1 (Wang et al, 2018). We also tried an additional network architecture, ResNet-101. Finally, we experimented with Black box attacks, following the standard setting, as in the paper you recommended (Moustafa et al, ICCV 2019). In all cases, we demonstrate superior performance and increased robustness to adversarial attacks.
>
> $\bullet \textit{- Insufficient experiment with larger datasets (such as CIFAR-100) or enough variety of datasets (such as SVHN).}$
>
> Thanks for your suggestion. We accordingly ran additional experiments with a ResNet-18 on CIFAR-100 dataset and in the table below we report the new results (classification accuracy, in percent). Note, that our approach produces consistently more robust models.
>
> One important observation is that our method is consistently robust to adversarial attacks, and in most cases generates robust models than employing adversarial training, while achieving higher accuracy on the clean data. As can be observed in the last three rows, our method can also be easily combined with adversarial training, in which case robustness is greatly improved.
>
> $\begin{array} {|r|r|}\hline Method & clean & FGSM(2/8/16) & BIM(2/8/16) & PGD(2/8/16) \\ \hline baseline & 76.2 & 21.8/8.7/4.5 & 11.0/0/0 & 11.6/0/0 \\ \hline \theta=0.95 & 68.9 & 65.8/57.6/47.9 & 56.6/40.6/37.6 & 57.5/36.7/26.6 \\ \hline \theta=0.9 & 67.2 & 64.5/56.5/48.9 & 55.7/44.1/41.8 & 58.3/38.6/29.0 \\ \hline \theta=0.8 & 62.6 & 58.8/54.2/48.3 & 54.4/44.7/43.0 & 55.3/41.8/32.0 \\ \hline adv. train & 61.7 & 47.4/23.1/11.0 & 47.2/17.4/4.0 & 48.0/20.4/5.5 \\ \hline \theta=0.95 + adv & 60.4 & 58.5/57.1/53.2 & 56.9/53.4/50.2 & 56.8/54.5/50.0 \\ \hline \theta=0.9 + adv & 58.6 & 58.5/56.2/54.2 & 58.0/53.6/52.6 & 57.4/53.8/49.8 \\ \hline \theta=0.8 + adv & 56.7 & 54.8/52.6/51.3 & 53.9/51.2/50.6 & 55.2/51/8/49.5 \\ \hline  \end{array}$
>
> $\bullet \textit{ - No direct experiment verification that supports the advantage of randomization in a subspace }$
> To further showcase the advantages of our approach against ones that don’t make the randomization in the subspace we compare against the method suggested by Reviewer 1 (Wang et al 2018), in which the authors proposed to apply dropout directly to the activation of the first fully connected layer at test time. Since their method requires the presence of multiple fully connected layers, we apply our randomized tensorization directly on their architecture. The results of the comparison can be seen below for the same epsilon as in Wang et al 2018. Notice that our approach consistently outperforms the defensive dropout.
>
> $\begin{array} {|r|r|}\hline Method & Clean & FGSM & BIM & PGD \\ \hline Ours & 85.9 & 60.0 & 42.6 & 43.8 \\ \hline Defensive~dropout~[Wang~et~al]
>  & 83.4 & 41.3 & 32.2 & 35.2 \\ \hline \end{array}$
>
> $\bullet \textit{- No discussions on the training complexities and the extendability to large-scale datasets/networks, such as ImageNet/ResNet-101. }$
>
> Our proposed method is architecture-agnostic and can be incorporated in any arbitrary network architecture. To validate this, we follow your suggestion to train a ResNet-101 on the CIFAR10 dataset. The results are in line with those obtained using a ResNet-18 and we report results from this additional experiment below:
>
> $\begin{array} {|r|r|}\hline Method & Clean & FGSM(2/8/16) & BIM(2/8/16) & PGD(2/8/16) \\ \hline baseline & 95.4 & 65.8/48.7/28.6 & 43.3/0/0 & 44.6/0/0 \\ \hline \theta=0.95 & 92.1 & 84.8/72.8/58.5 & 72.9/40.0/39.9 & 75.6/42.5/37.0 \\ \hline adv. train & 87.5 & 76.7/54.1/37.0 & 74.5/43.5/26.0 & 76.2/48.1/28.4 \\ \hline \theta=0.95+adv & 86.6 & 84.3/79.6/73.0 & 83.3/73.8/62.5 & 84.1/76.0/69.2 \\ \hline  \end{array}$
>
> Our method is significantly more robust to adversarial attacks than the baseline, and even outperforms adversarial training. Robustness can be further improved by combining the two.
>
> $\bullet \textit{- Missing citation and comparison to the following two SOTAs:}$
> $\textit{1. Xie et al., Feature Denoising for Improving Adversarial Robustness, CVPR19 }$
> $\textit{2. Mustafa et al., Adversarial Defense by Restricting the Hidden Space of Deep Neural Networks, ICCV19}$
>
> Thank you for pointing these out, we will cite them and add them to the discussion in the updated manuscript. The last one in particular was not published at the time of writing the initial version of our paper.

---

> > ### Author Response · Authors · 2019-11-14
> > **Response to Reviewer 2 [part 2/2]**
> >
> >
> > $\bullet \textit{About experiments against black box attacks}$
> >
> > Thank you for your comment. In addition to the additional experiments mentioned and reported above, we ran a black box attack. We followed the standard setting, as in the paper you recommended (Moustafa et al, ICCV 2019). The results are as follows:
> >
> > $\begin{array} {|r|r|}\hline Method & Clean & FGSM(2/8/16) & BIM(2/8/16) & PGD(2/8/16) \\ \hline Baseline & 95.4 & 94.2/87.8/79.1 & 93.0/84.7/77.5 & 94.0/82.3/59.9 \\ \hline Ours & 88.5 & 87.4/87.1/83.3 & 87.4/84.5/83.8 & 87.6/85.2/83.4 \\ \hline  \end{array}$
> >
> > As expected, our method is more robust in all cases. One thing to notice is the relative difference in performance: our method starts with a slightly lower performance on the clean set but is much less affected by the adversarial attacks.
> >
> > $\bullet \textit{It is unclear how much the architecture of a backbone can impact the fairness of comparison.}$
> > We agree with the reviewer that a fair comparison is paramount. As such, we compared only methods in the same context. In particular, all our comparisons are done on the same dataset, using the same exact architecture and the same experimental setting. This is also why it is challenging to compare with all methods, and we selected the most recent state-of-the-art at the time of writing to compare with. In case this is not sufficiently clear in the current version, we will further clarify this in the updated manuscript.
> >
> > $\bullet \textit{On typos and unifying the subscripts/superscripts for all \lambda.}$
> > Thank you for pointing these out, we will fix them in the updated manuscript.

---

### Public Comment · ~Anthony_Wittmer1 · 2019-09-28
**Evaluation questions and obfuscated gradients**

Hi,

I find the evaluation on the black-box attacks is missing in this paper, which is important, because that if a model causes obfuscated gradients, black-box attacks perform better than white-box attacks[1].

Some defend methods based on the randomization techniques have been broken by BPDA[1] or Nattack[2].
Since the proposed method adopts some randomization techniques, I have a little doubt whether the proposed model causes obfuscated gradients to give a false sense of security.

In order to check whether obfuscated gradients has happened, BPDA[1] or Nattack[2] is a better choice to evaluate the models.

[1] Obfuscated Gradients Give a False Sense of Security: Circumventing Defenses to Adversarial Examples. ICML 2018
[2] NATTACK: Learning the Distributions of Adversarial Examples for an Improved Black-Box Attack on Deep Neural Networks. ICML 2019

---

> ### Author Response · Authors · 2019-10-04
> **Additional comparison wtih BPDA [1]**
>
> Hi Anthony,
>
> Thank you for your interest in our paper.
>
> In Section 5, "Defending against omniscient attacker", we hoped to address that very point. In that scenario, the attacker has access to the full (unrandomized) weights and uses these to perform the attack. The idea is that these unrandomized weights could be obtained by accumulating forward passes as suggested in BPDA[1].
> Please note that an important point of our approach is that we do not randomize the weights directly. Instead, we apply randomization in the latent subspace spanned by the low-rank structure imposed. You can think of it as a stochastic regularization applied to the *rank* of the tensor factorization.
>
> One could argue that applying [1] would result in different results (e.g. [1] acts as an ensemble of models). To verify, in addition to the above scenario and following your comments, we ran the following additional experiment (BPDA [1]):
> at each iteration of gradient descent, for each convolutional layer, instead of taking a step in the direction of $\nabla_x f(x)$  we move in the direction of $\sum_{i=1}^{k}\nabla_x f(x)$ where each pass has the weights randomized in the latent space using our approach.
>
> We report here the accuracy (on CIFAR 10), obtained using our best model, for various values of $\epsilon$:
> +---------------+----------------------------+
> |                   |          Epsilon            |
> | Attack      +-------+---------+---------+
> |                   | 2      | 8        | 16      |
> +---------------+-------+--------+----------+
> | BPDA [1] | 83.3 | 54.9  | 43.8   |
> +---------------+-------+--------+----------+
>
>
>
> While in [1] the authors use $k=10$ we try with up to $k=20$ but without noticing any significant increase in the success rate of the attack. The PGD attack itself was run for 500 iterations as in [1]. These results are in line with the results we reported in the paper, see Table 2 in the manuscript.
>
> Thanks,
> The authors.

---

### Author Response · Authors · 2019-11-14
**General response to all reviewers**


We are glad to see that all reviewers recognised the novelty of our approach and that there is a consensus for acceptance. We are grateful to all reviewers’ comments, which we believe will help in greatly improving the quality of the paper. In this rebuttal, we carefully addressed all the comments and ran additional experiments as recommended by the reviewers within the limited time.

$\textbf{A summary of additional experiments conducted during rebuttal}$:
Following the reviewers’ suggestions, we ran the following additional experiments:
i) Experiment on CIFAR-100
ii) Experiments with a different, deeper architecture (ResNet-101)
iii) Experiments with BPDA [1] (test against obfuscated gradients)
iv) Comparison with matrix-based decomposition
v) Comparison with Wang et al, 2018.
vi) Comparison with defensive dropout
vii) Experiments with black box attack

We address individual reviewer’s comments in their respective threads.

---

### Decision · Program_Chairs · 2019-12-19

**Decision:**

Reject

**Comment:**

Three reviewers have assessed this submission and were moderately positive about it . However, the reviewers have also raised a number of concerns. Initially, they complained about substandard experimentation which has been resolved to some degree after rebuttal (rev. believe more can be done in terms of unifying them, investigating backbones, attack methods, and experimental settings in light of recent papers).

A somewhat bigger criticism concerns the theoretical part:
1. Rev. remained unclear why using tensor decomposition techniques is a sound approach for designing robust network.
2. AC and rev. also noted during discussions that using low rank constraints (and other mechanisms) and i.e. encouraging smoothness (one important mechanism among many in robustness to attacks) have been extensively investigated in the literature, yet, the proposed idea makes scarce if any theoretical connection to such important theoretical tools.

Some references (not exhaustive) that may help authors further study the above aspects are:
Certified Adversarial Robustness via Randomized Smoothing, Cohen et al.
Local Gradients Smoothing: Defense against localized adversarial attacks, Naseer et al.
Limitations of the Lipschitz constant as adefense against adversarial examples, Huster et al.
Learning Low-Rank Representations, Huster et al.

On balance, AC feels that despite the enthusiasm, this paper is not ready yet for the publication in ICLR as the key theory behind the proposed idea is missing. Thus, this submission falls marginally short of acceptance in ICLR 2020. However, the authors are encouraged to build up a compelling theory and resubmit to another venue (currently the paper feels like a solid workshop idea that needs to be investigated further).